# How much does it cost to measure immunity? A costing analysis of a measles and rubella serosurvey in southern Zambia

**Andrea C. Carcelen** [1] *, **Kyla Hayford**[1], **William J. Moss**[1,2], **Christopher Book**[3], **Philip E. Thuma**[3], **Francis D. Mwansa**[4], **Bryan Patenaude**[1]

**1** Department of International Health, Johns Hopkins Bloomberg School of Public Health, Baltimore, Maryland, United States of America, **2** Department of Epidemiology, Johns Hopkins Bloomberg School of Public Health, Baltimore, Maryland, United States of America, **3** Macha Research Trust, Choma, Zambia, **4** Ministry of Health, Lusaka, Zambia

* acarcel1@jh.edu

**Data Availability Statement:** All relevant data are within the paper and its Supporting Information files.

## Abstract

### Background

Serosurveys are a valuable surveillance tool because they provide a more direct measure of population immunity to infectious diseases, such as measles and rubella, than vaccination coverage estimates. However, there is concern that serological surveys are costly. We adapted a framework to capture the costs associated with conducting a serosurvey in Zambia.

### Methods

We costed a nested serosurvey in Southern Province, Zambia that collected dried blood spots from household residents in a post-campaign vaccine coverage survey. The financial costs were estimated using an ingredients-based costing approach. Inputs included personnel, transportation, field consumable items, social mobilization, laboratory supplies, and capital items, and were classified by serosurvey function (survey preparation, data collection, biospecimen collection, laboratory testing, and coordination). Inputs were stratified by whether they were applicable to surveys in general or attributable specifically to serosurveys. Finally, we calculated the average cost per cluster and participant.

### Results

We estimated the total nested serosurvey cost was US $68,558 to collect dried blood spots from 658 participants in one province in Zambia. A breakdown of the cost by serosurvey phase showed data collection accounted for almost one third of the total serosurvey cost (32%), followed by survey preparation (25%) and biospecimen collection (20%). Analysis by input categories indicated personnel costs were the largest contributing input to overall serosurvey costs (51%), transportation was second (23%), and field consumables were third (9%). By combining the serosurvey with a vaccination coverage survey, there was a savings of $43,957. We estimated it cost $4,285 per average cluster and $104 per average participant sampled.

**Funding:** The study was funded by the Bill & Melinda Gates Foundation (https://www.gatesfoundation.org/), grant number OPP1094816 awarded to KH and WJM as co-principal investigators. The funders had no role in study design, data collection and analysis, decision to publish, or preparation of the manuscript.

**Competing interests:** AC, KH, WM, and BP received financial support from the Bill & Melinda Gates Foundation. This does not alter our adherence to PLOS ONE policies on sharing data and materials. Other authors declare that they have no competing interests.

## Conclusions

Adding serological specimen collection to a planned vaccination coverage survey provided a more direct measurement of population immunity among a wide age group but increased the cost by approximately one-third. Future serosurveys could consider ways to leverage existing surveys conducted for other purposes to minimize costs.

## Introduction

Monitoring population immunity to measles and rubella viruses can help identify populations at risk of outbreaks and determine whether targeted vaccination efforts are needed. Vaccination coverage may be used to approximate population immunity levels but inaccuracies result because vaccinated individuals can remain susceptible and unvaccinated individuals can be immune following infection. Serosurveys provide a more direct measure of population immunity to infectious diseases such as measles and rubella [1]. The use of serosurveys to identify population immunity gaps to measles and rubella has increased globally with the establishment of regional measles and rubella elimination goals [2]. However, there is concern that serosurveys are costly and time consuming [3]. World Health Organization (WHO) measles and rubella serosurvey guidelines suggest serosurveys could require from $100,000 to over $1 million and require a one-year timeline to conduct, analyze and interpret [4]. Reducing costs could make serosurveys more feasible in low- and middle-income countries where fewer serosurveys have been conducted [5].

Some expenses would be required for any survey, regardless of whether they include biospecimen collection, such as vaccination coverage surveys or Demographic and Health Surveys [6]. These include data collection expenses for travel and fieldwork, including participant enrollment and questionnaire administration [7]. Another major driver of cost is the sample size, which is determined by the prevalence of what is being estimated and the desired precision [8]. Sampling strategies influence the cost of surveys and understanding the implications of different sampling strategies on cost could help determine the feasibility [9]. For example, the cost could be substantially reduced if biospecimens were collected from a subset of participants within a larger survey.

On the other hand, some costs are unique to serosurveys because they require biospecimens. Serosurveys can prospectively collect biospecimens or use specimens that have already been collected, such as a biorepository or residual samples. While prospective specimen collection allows for better control of data collection and sampling methodologies, collecting, processing and transporting blood specimens can be expensive and logistically challenging [3, 10]. Adding specimen collection to a planned survey could result in cost savings compared to a standalone serosurvey; however, there are concerns about logistical feasibility and cost [11, 12].

Biospecimen collection adds expenses and complexity to a survey in terms of human resources and supplies. Healthcare professionals or skilled workers trained in biospecimen collection are needed. For example, venous blood collection requires a phlebotomist or healthcare professional, whereas finger prick blood collection can be done by trained community health workers [13, 14]. Biospecimens also require specimen collection supplies, storage and transport, laboratory testing, and additional human resources for laboratory processing and testing, although the use of dried blood spots can reduce the costs of transport, processing and storage [11].

The aim of this study was to better understand the costs associated with a serosurvey and how costs could be minimized. We adapted an existing framework on the components and costs of integrated vaccine-preventable disease surveillance to the components of a serosurvey [15]. We populated the framework with costing data from a serosurvey conducted in Southern Province, Zambia to examine how different components of the serosurvey affected the total cost.

## Materials and methods

### Study location and design

This serosurvey was conducted in Southern Province, Zambia, where measles and rubella remain endemic [16]. Following a measles-rubella vaccination campaign in 2016 for children younger than 15 years of age, a national post-campaign vaccination coverage survey was conducted. In conjunction with this survey, a nested serosurvey was conducted that involved collecting biospecimens from all members of households enrolled in the post-campaign vaccine coverage survey, including adults. Dried blood spots were collected by finger prick for enrolled individuals [17]. The WHO vaccination coverage survey manual was used to guide the household-based survey design [18]. Clusters were defined as small geographic administrative boundaries, known as enumeration areas, based on the most recent census conducted in 2010. The vaccination coverage survey enrolled 12 children per cluster among those eligible for vaccination at the time of the campaign. Sixteen of the 26 clusters selected for the coverage survey in Southern Province were included in the serosurvey for logistical reasons.

### Cost data

The financial costs were estimated using an ingredients-based costing approach in which each resource is identified and assigned a cost. [19, 20]. The costing analysis was performed from the perspective of the government healthcare system and participant costs were not incorporated. In this case, a non-governmental organization was hired to implement the serosurvey. Cost data were captured in local currency (Zambian Kwacha) and converted to US dollars using the annual exchange rate in 2016 (USD $1 = 10.3 Kwacha) [21]. We considered a two-month time horizon for implementation of the serosurvey, from planning to laboratory processing. Weighting for incremental costs and discounting were not performed. Serosurvey cost data were obtained through document review of the budget and program records (such as purchase orders and contracts) and interviewing administrative personnel. When there was a difference between the budget and reported expenditure, reported expenditure was used to more accurately reflect the serosurvey as it was implemented rather than designed [20].

### Ethics statement

This study used financial costing information and did not involve human subjects research. However, the serosurvey upon which estimates are based was approved by Institutional Review Boards at Macha Research Trust (E2016.04) and the Johns Hopkins Bloomberg School of Public Health (00007447). Regulatory approval for this publication was granted by the National Health Research Authority in Zambia.

### Serosurvey cost estimation

A costing framework for integrated disease surveillance was adapted to capture the categories of implementation inputs across the phases of a serosurvey (Fig 1). Serosurvey phases were the activities required for implementation, from planning to biospecimen collection to laboratory testing. Cross-cutting items that spanned across serosurvey phases, such as communication,

**Fig 1. Framework for estimating serosurvey costs.** The framework was adapted from integrated disease surveillance and updated to capture serosurvey costs. Phase of study includes cores study activities. Columns represent input categories. Overlap across the matrix is captured in the costs.

supervision, and data management, were placed in a separate category of coordination. The framework was compared with Demographic and Health Surveys and vaccination coverage survey budgets to ensure all costs were captured [6]. For each category, we identified the proportion of costs attributed to a vaccination coverage survey and those specific to the serosurvey. Costs were stratified by whether they varied at the study, cluster, or participant level to allow calculation of marginal costs [22].

**Personnel.** We recorded the number of workers required for each activity. For long-term personnel who temporarily supported the serosurvey, proportioned salary and benefits were allocated based on the time spent doing serosurvey activities. Since this was a one-time intensified activity, two months were allocated for most existing personnel. For personnel contracted specifically for the serosurvey, all contracted time was included. We also included fees of consultants who supported specific services, such as training. We apportioned total personnel costs to serosurvey activities based on the ratio of time spent performing serosurvey-specific activities (e.g., blood collection) compared to general survey activities (e.g., mapping cluster). For the base case, one team was assumed to complete the serosurvey in one cluster within three days. The team consisted of a supervisor, phlebotomist, and three data collectors.

**Transportation.** No vehicles were purchased for the serosurvey; however, vehicles were rented at a fixed per day cost for vehicle use and driver time. Most transportation costs were allocated to general survey activities. Only trips to collect and transport biospecimens to the lab were included in serosurvey-specific costs. Each team was assigned one vehicle to implement field activities for the serosurvey.

**Field consumables.** Field consumables included items required during data and biospecimen collection and were stratified by items required for general survey implementation, such as pens, and serosurvey-specific items, such as biospecimen collection kits (e.g. lancets, cotton swabs, gloves). General survey items typically were calculated per cluster as they were team-level costs. Serosurvey specific items were typically calculated per participant.

**Social mobilization.** Because social mobilization for the post-campaign vaccine coverage survey was done as part of the vaccination campaign, all additional social mobilization efforts were calculated as serosurvey-specific costs. These included stipends for community health workers to accompany the teams in the field during data collection, radio advertisements, and meetings and phone calls with health facility staff to notify them that a serosurvey was being conducted in their area.

**Laboratory supplies.** We assumed the laboratory providing services for a serosurvey had the equipment to conduct enzyme immunoassays using commercial kits. The cost of bench space was based on renting the laboratory space and equipment for the time required for laboratory testing. Laboratory costs were calculated based on the cost of supplies and consumable materials (e.g. test kits, gloves, tubes). It was assumed that 15% of biospecimens would be retested in the base case to account for quality assurance and quality control, as well as retesting biospecimens with equivocal results and biospecimens with results above the upper limit of detection for a commercial measles and rubella IgG enzyme immunoassay kit.

**Capital items and overhead.** Because we costed a single serosurvey, the only capital items purchased specifically for the serosurvey were tablets for data entry. All other equipment was borrowed, and use-time was included in the cost based on the serosurvey implementation time with no discounting. No overhead costs were included.

## Data analysis

We used Microsoft Excel to compile and analyze the data. Using the number of clusters and participants included in the serosurvey, we calculated the average per cluster and per participant costs, as well as the marginal cost per cluster and participant. Estimated costs were then stratified by serosurvey activity. Inputs were also stratified by whether they were general to surveys or attributable specifically to serosurveys. One-way sensitivity analyses were conducted for personnel and time spent on serosurvey activities.

Marginal costs for an additional cluster were calculated based on social mobilization, personnel, survey materials, and transportation required to include a cluster in the survey. This included one day for enumeration, mapping, and social mobilization preparatory activities required at the cluster. Marginal costs for an additional participant in an existing cluster included consumables for biospecimen collection and laboratory testing as well as additional time for personnel and transport.

## Results

Biospecimens were collected from 658 individuals in 16 clusters by four teams comprised of five members each (supervisor, interviewers, and blood collector). Each team took approximately three days to complete each cluster. One and a half weeks were allocated for training and piloting, and three weeks for laboratory testing. With these assumptions, we estimated a cost of US $68,558 to collect biospecimens within a household vaccination coverage survey in Southern Province, Zambia. This resulted in a cost of $4,285 per average cluster and $104 per average participant sampled.

The overall added cost of collecting and testing a biospecimen as part of the survey was $24,601. Broken down by serosurvey phase, data collection accounted for almost one third of the total serosurvey cost (32%), followed by survey preparation (25%) and biospecimen collection (20%) (Table 1). By input categories, personnel was the largest contributing input to overall serosurvey cost (51%), transportation was second (23%), and field consumables were third (9%) (Fig 2). In terms of costs attributable exclusively to serosurveys, personnel was also the largest input (34%) due to the additional time required for data and specimen collection plus

**Table 1. Costs per phase of serosurvey implementation in Southern Province, Zambia.**

| Serosurvey phase | Cost (2016 USD) | Percentage of total cost |
|---|---|---|
| Survey preparation | $ 16,813 | 25% |
| Data collection | $ 22,062 | 32% |
| Biospecimen collection | $ 13,875 | 20% |
| Laboratory testing | $ 10,726 | 16% |
| Coordination (Communication, Data management) | $ 5,081 | 7% |
| TOTAL | $ 68,558 | 100% |

Phases of the serosurvey correspond to Fig 1. Percentages sum to 100%. All costs in 2016 USD.

the addition of laboratory personnel, followed by laboratory supplies (26%) and social mobilization (18%). By combining the serosurvey with the vaccination coverage survey, there was a savings of $43,957, the cost attributable to the survey not related to biospecimens.

We estimated that the marginal cost of including an additional cluster was $1,620, and the marginal cost of adding a participant within an existing cluster was $30. For these, 46% of the cost to add a cluster and 64% of the cost to add a participant were attributable to serosurvey-specific costs. Additional information on the implications of cluster and participant costs can be found in the S1 File.

In the serosurvey, all members of the household, including adults as well as children, were enrolled, whereas only children were included in the vaccination coverage survey. The sample size for the nested serosurvey required only 192 children in the 16 clusters (S1 File). By incorporating more than one child per household, resulting in an additional 235 children, we added $7,030 to the serosurvey cost. Including 219 adults in the households included in the serosurvey cost an additional $6,552.

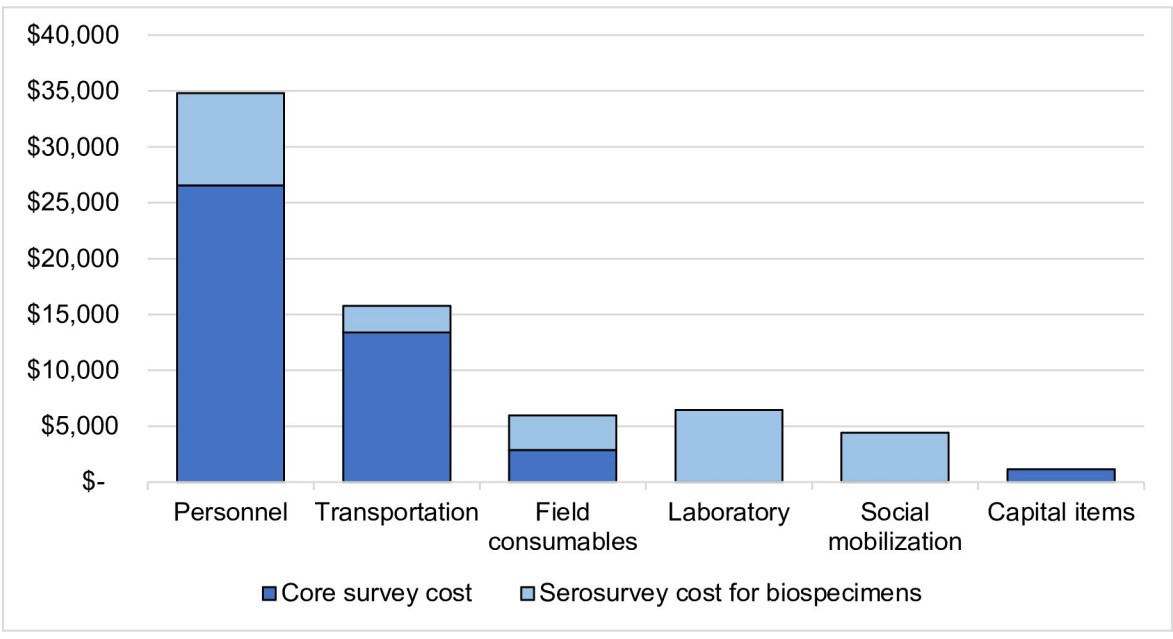

**Fig 2. Costs for the post-campaign vaccine coverage survey and serosurvey in Southern Province, Zambia by input category.** Costs captured for each input category span across serosurvey activities. Dark blue represents core survey costs that would be included in a vaccination coverage survey, while light blue represents the costs that are specifically attributable to a serosurvey due to blood specimen collection and testing, such as laboratory costs. All bars sum to the total cost of $68,558. All costs are in 2016 USD.

**Table 2. Costs per susceptible participant identified in 2016 serosurvey in Southern Province, Zambia.**

| Serosurvey phase | No. persons susceptible | Cost for identifying a susceptible person |
|---|---|---|
| Measles | 33 | $ 2,077 |
| Rubella | 18 | $ 3,809 |
| Combined | 51 | $ 1,344 |

Few participants were seronegative for measles or rubella, as the serosurvey was conducted after an immunization campaign [17]. The average cost per person found to be seronegative to measles was $2,077, and $3,809 for rubella (Table 2). Accounting for both measles and rubella, the cost was $1,344 per susceptible person identified.

Sensitivity analyses showed that varying the time for field work and number of team members had the largest impact on serosurvey cost, as this affected field work time (Fig 3). These factors impacted the data collection phase, which accounted for most of the study cost. However, none of the sensitivity analyses changed the cost by more than 8%.

## Discussion

We estimated the total nested serosurvey cost was US $68,558 to collect dried blood spots from 658 participants in one province in Zambia and that nesting the serosurvey in a vaccination coverage survey added $24,600 in overall costs. A breakdown of the cost by serosurvey phase showed data collection accounted for almost one third of the total serosurvey cost (32%), followed by survey preparation (25%) and biospecimen collection (20%). Analysis by input categories indicated personnel was the largest contributing input to the overall serosurvey cost (51%), transportation was second (23%), and field consumables were third (9%). By combining the serosurvey with the vaccination coverage survey, there was a savings of $43,957. We estimated it cost $4,285 per average cluster and $104 per average participant sampled.

Few costing studies of serological surveillance have been conducted, and to our knowledge, none have been done for measles and rubella serosurveys. Economic evaluations of public health surveillance systems have used various methodologies, making it difficult to compare across settings [23]. We developed a framework for serological surveillance that can be used to

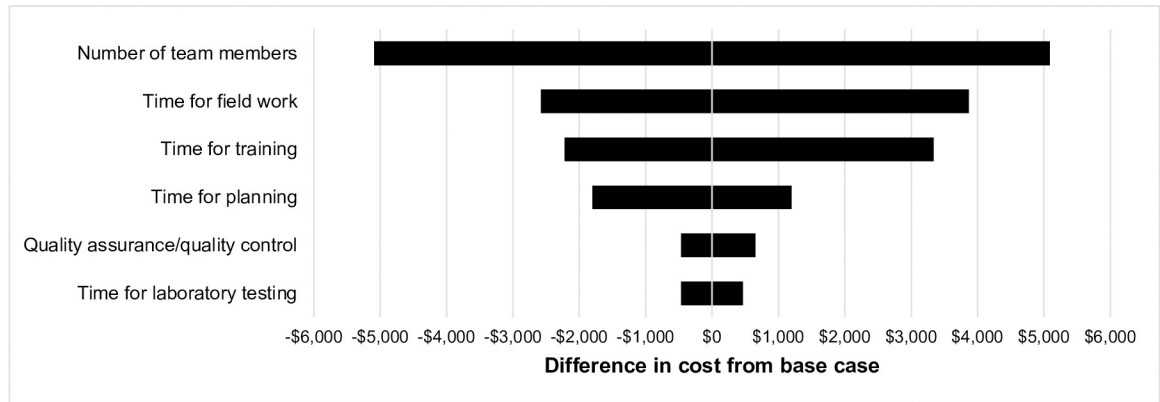

**Fig 3. One-way sensitivity analyses of costs varying serosurvey parameters.** One-way sensitivity analysis representing how varying parameters changed the total serosurvey cost from the base case of $68,558. Negative values (to the left) indicate lower cost than the base case, reflecting the lower end of the ranges, and the positive values (to the right) indicate higher cost than the base case, reflecting the higher end of the ranges. The number of team members varied from 2 to 6, the time for field work varied from 10 to 15 days, the time for training varied from 3 to 8 days, and the time for planning varied from 5 to 10 days. The percentage of specimens requiring retesting varied from 5 to 25% and the time for laboratory testing varied from 13 to 23 days. All costs are in 2016 USD.

compare serosurvey costs. Standardized categories permit cross-country comparisons and could be used as a model for other countries considering serological surveillance.

WHO estimated that 60 to 70% of a serosurvey budget is laboratory-related supplies for blood collection, storage, transport, processing, and testing kits [2]. We estimated that the added time for biospecimen collection in the field and laboratory-related supplies were only approximately one-third of the overall serosurvey budget. This could be due to the limited serosurvey transportation cost as we did not have vehicles allocated specifically for biospecimen transport. Another potential explanation for these differences could be the biological specimen type. Since we used finger prick blood collection, this may have reduced the costs included in biospecimen collection as it does not require personnel skilled in venous blood draw. Collecting biospecimens as dried blood spots eliminated the need for a cold chain and time-sensitive transportation to the laboratory.

The additional cost for the serosurvey in Zambia found seroprevalence was higher than vaccination coverage reported in the survey for the measles-rubella vaccination campaign in 2016. Measles seroprevalence was 96.1% (95% CI: 92.4, 98.1), and rubella was 98.4% (95% CI: 95.9, 99.4) for children 9 months to 16 years of age. By comparison, the vaccination coverage was only 89.9% (95% confidence interval (CI): 85.9, 92.8) [17]. For measles, 95% immunity is considered a herd immunity threshold to interrupt virus transmission [24]. Adding the serosurvey demonstrated that this programmatic goal was reached, despite the supplemental cost and logistics required for the serosurvey.

Although the MR vaccination campaign targeted children younger than 15 years of age, monitoring seroprevalence in adults is important because, as measles and rubella virus transmission diminishes, fewer people are immunized through natural infection in childhood and will be at risk of acquiring disease as adults. This is of particular concern for women of childbearing age as it could result in increased risk of congenital rubella syndrome [25]. In this serosurvey, lower rubella seroprevalence was identified in women of childbearing age [17]. It cost an additional $6,552 to include adults in the serosurvey, which revealed immunity gaps among young adults not eligible for the campaign. These gaps would not have been identified through the vaccination coverage survey alone. This adult population can be monitored through serological surveillance without a substantial increase in resources.

Understanding factors that have the greatest impact on serosurvey cost can be used to minimize these expenses. Estimates that only consider laboratory supplies for testing biospecimens underestimate the additional costs for specimen collection, such as personnel and transportation. Other household surveys have noted that the cost of survey implementation in sub-Saharan Africa often has high personnel costs [26]. An immunization program costing study in Zambia also identified personnel and travel as the highest implementation costs for routine immunization [27]. Similarly, our survey identified personnel and transportation as the highest cost inputs; however, we were able to save $26,539 in personnel and $13,410 in transportation by nesting within the post-campaign vaccine coverage survey.

Alternative study designs, such as using an existing biorepository rather than collecting new biospecimens, could achieve additional cost-savings in data and biospecimen collection. Together these categories accounted for more than half the cost of the serosurvey. Other ways to reduce costs include limiting training time by using experienced data collectors and improving data collection tools to minimize time in the field.

As new technologies continue to develop, such as point-of-care serological tests, laboratory costs could be reduced [28, 29]. These technologies would not require specimen transportation and could be done by non-laboratory personnel, thereby decreasing costs [30]. The use of multiplex bead-based assays to detect multiple antigens from the same biospecimen could make serosurveys more cost-effective by providing information on multiple diseases in less time without requiring additional biospecimen collection [31].

It is not yet clear what is the appropriate indicator to weigh the costs and benefits of a serosurvey. We estimated the cost to identify someone who is seronegative, which was high in this population because of the high seroprevalence. The cost per seronegative individual will vary across settings and would be lower in settings with low seroprevalence. Weighing the value of information gained from a serosurvey could help develop more appropriate benefit estimates.

## Limitations

Data for this study were collected retrospectively, so some of these limitations could be addressed if using a prospective study design. Transportation costs did not capture actual distances covered. Capital laboratory equipment was not individually valued and annualized, but rather a lump sum for laboratory bench space was used. We were not able to estimate the marginal cost per household due to the format of the costing data and the variable number of people included per household; therefore, estimates were based on participant costs. These costing estimates were for the number of participants enrolled and may not account for additional time required if conducting a serosurvey in a setting with high refusal or non-response rates that would require additional time spent on enrollment. Because this was a one-time activity, new personnel were not hired but contracted for their time. If this were to be an ongoing activity, hiring additional personnel or allocating a proportion of existing personnel time could decrease costs.

## Conclusions

Adding serological specimen collection to a planned vaccination coverage survey in Southern Province, Zambia provided a more direct measurement of population immunity while increasing the cost by approximately one-third. By nesting this serosurvey within a planned post-campaign vaccine coverage survey, costs for planning, personnel, mapping, enumeration, and transportation were not borne by the serosurvey. Despite these savings, personnel and laboratory supplies remain significant drivers of cost. Future serosurveys could consider ways to leverage existing surveys for other purposes to minimize costs.

## Supporting information

**S1 File. Serosurvey sample size calculations and cost ratio of cluster to participant cost.**
(DOCX)

**S1 Dataset.**
(XLSX)

## Acknowledgments

We thank the Government of Zambia, particularly the team that implemented the vaccination coverage survey, and Dr. Philip Thuma at Macha Research Trust for supporting the serosurvey which allowed us to capture programmatic costs. We would also like to thank Christine Prosperi and Simon Mutembo for their input and review.

## Author Contributions

**Conceptualization:** Andrea C. Carcelen, Kyla Hayford, William J. Moss.

**Data curation:** Andrea C. Carcelen, Christopher Book, Philip E. Thuma.

**Formal analysis:** Andrea C. Carcelen, Kyla Hayford, Bryan Patenaude.

**Funding acquisition:** Kyla Hayford, William J. Moss.

**Investigation:** Andrea C. Carcelen.

**Methodology:** Andrea C. Carcelen, Bryan Patenaude.

**Project administration:** Andrea C. Carcelen, Philip E. Thuma, Francis D. Mwansa.

**Resources:** William J. Moss.

**Software:** Andrea C. Carcelen.

**Supervision:** Andrea C. Carcelen, Kyla Hayford.

**Validation:** Bryan Patenaude.

**Visualization:** Andrea C. Carcelen.

**Writing – original draft:** Andrea C. Carcelen.

**Writing – review & editing:** Kyla Hayford, William J. Moss, Bryan Patenaude.

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
