## [Decision Letter · Decision Letter 0]

25 Jun 2020

PONE-D-20-11421

How much does it cost to measure immunity? A costing analysis of a measles and rubella serosurvey in southern Zambia

PLOS ONE

Dear Dr. Carcelen,

Thank you for submitting your manuscript to PLOS ONE. After careful consideration, we feel that it has merit but does not fully meet PLOS ONE’s publication criteria as it currently stands. Therefore, we invite you to submit a revised version of the manuscript that addresses the points raised during the review process.

ACADEMIC EDITOR: Please address the comments from the reviewers.

We look forward to receiving your revised manuscript.

Kind regards,

Ka Chun Chong

Academic Editor

PLOS ONE

Journal Requirements:

'AC, KH, WM, and BP received financial support from the Bill & Melinda Gates Foundation. Other authors declare that they have no competing interests'

Additional Editor Comments (if provided):

Reviewers' comments:

Reviewer's Responses to Questions

**Comments to the Author**

1. Is the manuscript technically sound, and do the data support the conclusions?

Reviewer #1: Yes

Reviewer #2: Yes

2. Has the statistical analysis been performed appropriately and rigorously? 

Reviewer #1: Yes

Reviewer #2: Yes

3. Have the authors made all data underlying the findings in their manuscript fully available?

Reviewer #1: No

Reviewer #2: Yes

4. Is the manuscript presented in an intelligible fashion and written in standard English?

Reviewer #1: No

Reviewer #2: Yes

5. Review Comments to the Author

Reviewer #1: Thank you for giving me the opportunity to review this interesting paper. I think the topic is relevant and based on sound data. However, I have several concerns regarding the presentation.

Abstract:

Background: the reason why serosurveys are valuable is not clearly stated.

Methods are more detailed than required

Results: Please see my comment to Results, line 214 to 223.

Introduction: The introduction is hard to read as it seems to be a collection of single one-sentence statements that lack context. Thus, the reason why the study was performed remains somewhat unclear. The aim of the study should be described better. To me, the phrase “cost integrated vaccine-preventable disease surveillance that includes calculating serosurvey” is not understandable.

Methods:

Line 105 to 106: “Clusters were defined as administrative boundaries known as Supervisory Enumeration Area” – what does that mean?

Line 112-115: Meaning unclear.

Line 133 to 137: Meaning unclear.

Line 155-57: Surprisingly high amount of Man-power to collect specimens, 3 working days à five persons to collect 41 specimens. Please explain why teams contain five persons.

Line 174.175: Why were community health care workers invited to accompany the working team, thus enhancing the costs?

Results:

Lien 214 to 223: I don’t understand the presentation of the percentages. In total 220 % are presented. How are they broken down?

Line 237 to 239: I do not see the relevance of this information.

Line 240 to 242: Incomplete sentence.

Line 242-243: Discussion of results

Line 244 to 250: Very detailed information, but unclear relevance.

Sensitivity analysis: I do not understand how the sensitivity analysis was performed.

Discussion:

No overview of the main results

Relevance of the study: It is not discussed how measuring seronegativity contributes to public health, or how the costs for a serosurvey relate to cost of other measurements to enhance public health, like vaccination campaigns or nutrition supplements.

Line 326 to 327: Incomplete& grammatically wrong sentence.

Line 330: not included into the pdf

Line 334 to 337: Unclear why this is a weakness

Acknowledgement: One of the co-authors is acknowledged for his contribution.

Figures: The figures are of insufficient quality in the pdf-file provided.

Reviewer #2: Thanks for the manuscript and it read very well. Please see the following comments for further improvement of the manuscript.

1. Line 244 to 248 should be in methods section for better clarity including the age range for serology survey.

2. Line 251 to 256 would be clear if the author presents as a table in result section.

3. The authors stated in the method section that 12 children were selected from each cluster. But in line 293 to 295 stated the fact about women of child bearing age which do not seem to be in this survey. Please clarify this fact.

4. It would be more informative if author could provide the administration of vaccination coverage in the survey area to argue with needs to conduct serology survey.

5. Figures in the manuscript is not clear and suggest to provide higher resolution figure.

6. PLOS authors have the option to publish the peer review history of their article (what does this mean?). If published, this will include your full peer review and any attached files.

Reviewer #1: No

Reviewer #2: No

---

## [Author Response · Author response to Decision Letter 0]

23 Aug 2020

Thank you kindly for the thorough review of the manuscript. We have reviewed the comments and provided responses below in blue. Additionally, we have updated the manuscript, and noted where in the manuscript the changes are reflected. Line numbers are based on unmarked version.

This has been updated.

The minimal anonymized data set necessary to replicate our study findings has been uploaded as supporting files.

'AC, KH, WM, and BP received financial support from the Bill & Melinda Gates Foundation. Other authors declare that they have no competing interests'

This is correct. It did not alter our adherence to PLOS ONE policies on sharing data and materials. We added this to the cover letter.

Reviewer #1: Thank you for giving me the opportunity to review this interesting paper. I think the topic is relevant and based on sound data. However, I have several concerns regarding the presentation.

Abstract:

Background: the reason why serosurveys are valuable is not clearly stated.

This has been added (line 21).

Methods are more detailed than required

Some of the details have been removed (lines 27-28).

Results: Please see my comment to Results, line 214 to 223.

This has been clarified (lines 34-39).

Introduction: The introduction is hard to read as it seems to be a collection of single one-sentence statements that lack context. Thus, the reason why the study was performed remains somewhat unclear. The aim of the study should be described better. To me, the phrase “cost integrated vaccine-preventable disease surveillance that includes calculating serosurvey” is not understandable.

The introduction has been reworked to be more cohesive. The aim of the study is more clearly described in the last paragraph, with the clarification on the phrase noted (lines 85-90).

Methods:

Line 105 to 106: “Clusters were defined as administrative boundaries known as Supervisory Enumeration Area” – what does that mean?

An Enumeration Area (EA) is a small geographic unit below a ward in urban areas and village in rural areas, into which the whole country is divided for census purposes. A supervisor is responsible for accounting for all of the people within their assigned enumeration areas. We tried to clarify that it is a small geographic area. (lines 100-102)

Line 112-115: Meaning unclear.

This has been modified. The intent was to explain that a bottom-up approach was used to cost each item required for the serosurvey. We included costs that were incurred by the government healthcare system as they implemented the serosurvey but did not include participant costs. (lines 107-110)

Line 133 to 137: Meaning unclear.

The text has been updated to be clearer. The intent was to explain Figure 1 and how the phases and categories were developed. (lines 127-132)

Line 155-57: Surprisingly high amount of Man-power to collect specimens, 3 working days à five persons to collect 41 specimens. Please explain why teams contain five persons.

The teams consisted of 5 people-1 supervisor, 1 phlebotomist, and 3 data collectors (one who was a back-up phlebotomist). This is the actual number of people that went out to the field for the serosurvey. Because this serosurvey was nested within a vaccination coverage survey, the teams conducting the serosurvey were asked to work at the same pace as those doing only the vaccination coverage survey. Whereas teams doing only the vaccination coverage survey in other clusters only needed to interview mothers of 12 children; serosurvey teams had to both interview and collect blood from everyone in the household of those 12 children. The mean number of participants enrolled in each household was 4 (IQR: 3-6) [1]. This meant that additional staff were required to complete all of the work within the allotted 2 weeks in the field. If more time had been allowed, the teams could have been smaller. 

Line 174.175: Why were community health care workers invited to accompany the working team, thus enhancing the costs?

Community health workers accompanied the teams for several reasons. They are familiar with the area and could guide the teams in interpreting the maps. And they could make appropriate introductions to community leaders so the community was more likely to participate in the study. [2] (Lines 167-171)

Results:

Lien 214 to 223: I don’t understand the presentation of the percentages. In total 220 % are presented. How are they broken down?

The paragraph has been reorganized to better clarify. The percentages are presented in a couple of formats: by phase of serosurvey implementation (as broken down in Table 1) and by input category (as denoted in Figure 2). This corresponds with the rows and columns from Figure 1. Additionally, lines 214-217 explain the costs per category input for just adding the serosurvey to the already planned coverage survey (the different colors in Figure 2). (Lines 209-218)

Line 237 to 239: I do not see the relevance of this information.

Cost ratio information has been moved to supplemental material.

Line 240 to 242: Incomplete sentence.

This sentence has been split into two and moved to supplemental material.

Line 242-243: Discussion of results

Cost ratio information has been moved to supplemental material.

Line 244 to 250: Very detailed information, but unclear relevance.

This has been condensed to get across the main points more clearly. The relevance was intended to denote that by adding additional children in the serosurvey than were in the vaccination coverage survey, there was an additional cost. Similarly, we included adults in the serosurvey sample, which also had an additional cost. The implications of this are in the discussion section (lines 295-304), highlighting that an immunity gap in women of childbearing age was identified through this serosurvey because adults were included.

Sensitivity analysis: I do not understand how the sensitivity analysis was performed.

One-way sensitivity analyses were done to see how changing parameters for the serosurvey would have affected the cost. For example, decreasing the number of team members to 2 resulted in a $5,000 cost savings, whereas increasing the number of team members to 6 resulted in increased costs. This was similarly done for time spent in the field, training, planning, and laboratory testing. The footnote for Figure 3 indicates what the ranges for the sensitivity analyses were. (lines 252-258)

Discussion:

No overview of the main results

An initial paragraph has been added to provide an overview and the value of serosurveys in guiding immunization programs (lines 261-269).

Relevance of the study: It is not discussed how measuring seronegativity contributes to public health, or how the costs for a serosurvey relate to cost of other measurements to enhance public health, like vaccination campaigns or nutrition supplements.

We agree this is a really important question, but unfortunately we were not able to address it in this study. We are working through our research to better understand the added value of targeted serosurveys and how they could be used to guide immunization programs. In future research we would do a cost-benefit or cost-utility analysis to be able to compare the costs of a serosurvey to those of alternative strategies. Alternative strategies could include approximate measurements of seropositivity using vaccination coverage and fever rash surveillance systems,. This could also compare a targeted serosurvey with mop-up immunization in areas found to have low seroprevalence to a blanket vaccination campaign (lines 325-329).

Line 326 to 327: Incomplete& grammatically wrong sentence.

Thank you for catching this. It has been updated.

Line 330: not included into the pdf

We are not sure what this is in reference to, but the minimal dataset has now been made available.

Line 334 to 337: Unclear why this is a weakness

It was not intended to be a weakness, but rather a limitation in terms of generalizability. In our study, we hired new personnel. If this were integrated as an ongoing activity that forms part of the existing surveillance system, it could be done by existing staff. This could result in lowered costs and would make it more sustainable in the long run, rather than hiring and training new personnel. (Lines 396-399)

Acknowledgement: One of the co-authors is acknowledged for his contribution.

This has been removed and updated (line 352).

Figures: The figures are of insufficient quality in the pdf-file provided.

These have been updated.

Reviewer #2: Thanks for the manuscript and it read very well. Please see the following comments for further improvement of the manuscript.

1. Line 244 to 248 should be in methods section for better clarity including the age range for serology survey.

This line is intended to be a reminder for the reader as to why we are saying “additional” children and what the cost was for adding adults. We have tried to make this clearer in the methods section. (lines 97-98)

2. Line 251 to 256 would be clear if the author presents as a table in result section.

A table has been added to the results section (line 244-246).

3. The authors stated in the method section that 12 children were selected from each cluster. But in line 293 to 295 stated the fact about women of child bearing age which do not seem to be in this survey. Please clarify this fact.

The post-coverage evaluation survey selected one child from 12 different households in each cluster. However, our nested serosurvey collected specimens from all household members regardless of age, not just the child. Therefore, we were able to calculate seroprevalence estimates for women of childbearing age.

4. It would be more informative if author could provide the administration of vaccination coverage in the survey area to argue with needs to conduct serology survey.

Administrative vaccination coverage for the 2016 measles-rubella vaccination campaign was 89.9% (95% confidence interval (CI): 85.9, 92.8) for 9 month to 16 year olds. By comparison, the serosurvey prevalence was 96.1% (95% CI: 92.4, 98.1) for measles and 98.4% (95% CI: 95.9, 99.4) for rubella for the same age range. For measles, the higher seroprevalence was likely due to the fact that many children had already been vaccinated through the routine immunization program. For rubella, this is likely because they had already been previously exposed to circulating rubella virus, as this was the first introduction of rubella vaccine. This justification for conducting serological surveillance has been added to the discussion (lines 286-294)

5. Figures in the manuscript is not clear and suggest to provide higher resolution figure.

These have been updated.

---

## [Decision Letter · Decision Letter 1]

14 Sep 2020

PONE-D-20-11421R1

How much does it cost to measure immunity? A costing analysis of a measles and rubella serosurvey in southern Zambia

PLOS ONE

Dear Dr. Carcelen,

Thank you for submitting your manuscript to PLOS ONE. After careful consideration, we feel that it has merit but does not fully meet PLOS ONE’s publication criteria as it currently stands. Therefore, we invite you to submit a revised version of the manuscript that addresses the points raised during the review process.

We look forward to receiving your revised manuscript.

Kind regards,

Ka Chun Chong

Academic Editor

PLOS ONE

Reviewers' comments:

Reviewer's Responses to Questions

**Comments to the Author**

1. If the authors have adequately addressed your comments raised in a previous round of review and you feel that this manuscript is now acceptable for publication, you may indicate that here to bypass the “Comments to the Author” section, enter your conflict of interest statement in the “Confidential to Editor” section, and submit your "Accept" recommendation.

Reviewer #1: (No Response)

Reviewer #2: All comments have been addressed

2. Is the manuscript technically sound, and do the data support the conclusions?

Reviewer #1: Yes

Reviewer #2: Yes

3. Has the statistical analysis been performed appropriately and rigorously? 

Reviewer #1: Yes

Reviewer #2: Yes

4. Have the authors made all data underlying the findings in their manuscript fully available?

Reviewer #1: Yes

Reviewer #2: Yes

5. Is the manuscript presented in an intelligible fashion and written in standard English?

Reviewer #1: No

Reviewer #2: Yes

6. Review Comments to the Author

Reviewer #1: Thank you for giving med the opportunity to re-revise this manuscript which I consider much improved.

My only comment is that there are a few typos, especially in the Introduction ("cost" instead of "costs" etc.), and iterations throughout the paper. I would recommend a critical run-through or a professional proof-read.

Reviewer #2: (No Response)

7. PLOS authors have the option to publish the peer review history of their article (what does this mean?). If published, this will include your full peer review and any attached files.

Reviewer #1: **Yes: **Corinna Vossius MD PhD

Reviewer #2: No

---

## [Author Response · Author response to Decision Letter 1]

25 Sep 2020

We have had several proof-readers go through and critically review the language. We found only a couple of typos but tried to update some of the language to prevent confusion between “cost” and “costs”.

---

## [Editor Report · Decision Letter 2]

2 Oct 2020

How much does it cost to measure immunity? A costing analysis of a measles and rubella serosurvey in southern Zambia

PONE-D-20-11421R2

Dear Dr. Carcelen,

We’re pleased to inform you that your manuscript has been judged scientifically suitable for publication and will be formally accepted for publication once it meets all outstanding technical requirements.

Kind regards,

Ka Chun Chong

Academic Editor

PLOS ONE
---

## [Editor Report · Acceptance letter]

6 Oct 2020

PONE-D-20-11421R2 

How much does it cost to measure immunity? A costing analysis of a measles and rubella serosurvey in southern Zambia 

Dear Dr. Carcelen:

I'm pleased to inform you that your manuscript has been deemed suitable for publication in PLOS ONE. Congratulations! Your manuscript is now with our production department. 

Kind regards, 

on behalf of

Dr. Ka Chun Chong 

Academic Editor

PLOS ONE